# Anti-RuvBL1/2 Autoantibodies Detection in a Patient with Overlap Systemic Sclerosis and Polymyositis

**DOI:** 10.3390/antib12010013

**Published:** 2023-02-03

**Authors:** Linda Di Pietro, Fabio Chiccoli, Lorenzo Salvati, Emanuele Vivarelli, Alessandra Vultaggio, Andrea Matucci, Chelsea Bentow, Michael Mahler, Paola Parronchi, Boaz Palterer

**Affiliations:** 1Department of Experimental and Clinical Medicine, University of Florence, Largo Brambilla, 3, 50121 Firenze, Italy; 2Allergology and Clinical Immunology Unit, Azienda Usl Toscana Sud Est, San Donato Hospital, 52100 Arezzo, Italy; 3Immunoallergology Unit, Careggi University Hospital, 50134 Firenze, Italy; 4Research and Development, Autoimmunity, Werfen, Autoimmunity Headquarters and Technology Center, San Diego, CA 92131-1638, USA; 5Immunology and Cell Therapy Unit, Careggi University Hospital, 50134 Firenze, Italy

**Keywords:** RuvBL1/2, autoantibodies, idiopathic inflammatory myopathy, polymyositis, speckled pattern, systemic sclerosis, particle multiple analyte technology, PMAT, scleromyositis

## Abstract

Anti-RuvBL1/2 autoantibodies have recently been detected in patients with systemic sclerosis (SSc) and scleromyositis overlap syndromes. These autoantibodies exhibit a distinct speckled pattern in an indirect immunofluorescent assay on Hep-2 cells. We report the case of a 48 year old man with facial changes, Raynaud’s phenomenon, puffy fingers, and muscle pain. A speckled pattern on Hep-2 cells was identified, but the conventional antibody testing was negative. Based on the clinical suspicion and the ANA pattern, further testing was sought demonstrating anti-RuvBL1/2 autoantibodies. Hence, a review of the English literature was performed to define this newly emerging clinical–serological syndrome. With the one here reported, a total of 52 cases have been described to date (December 2022). Anti-RuvBL1/2 autoantibodies are highly specific for SSc and are associated with SSc/PM overlaps. Apart from myopathy, gastrointestinal and pulmonary involvement are frequently observed in these patients (94% and 88%, respectively).

## 1. Introduction

Systemic sclerosis (SSc) is a connective tissue disease (CTD) characterized by fibroinflammatory phenomena and vasculopathy, involving the skin and internal organs. Despite some common features, such as Raynaud’s phenomenon (RP), the clinical presentations, epidemiological associations, and disease progression are quite heterogeneous. Historically, SSc was divided into limited and diffused forms, according to the skin involvement. It was soon observed how limited SSc (lSSc) affects predominantly the distal extremities and face and is associated with pulmonary hypertension and a relatively slow disease progression. The clinical constellation was described as calcinosis, RP, esophageal dysmotility, sclerodactyly, and telangiectasia (CREST) [1]. On the other hand, diffuse SSc (dSSc) was associated with rapidly progressive proximal and truncal skin involvement, digital ulcers, tendon friction rubs, interstitial lung disease (ILD), and scleroderma renal crisis. This marked clinical distinction was validated by the discovery that lSSc was strongly associated to anticentromere autoantibodies (ACAs), while dSSc with anti-topoisomerase I autoantibodies (ATAs), also known as anti-Scl-70 autoantibodies [2]. ACA and ATA were included in the classification criteria for SSc, and arguably, this was among the first definitions of clinical–serological entities.

Patients with SSc are virtually all positive for antinuclear antibodies (ANAs). However, around one-third of SSc patients still results “seronegative” if only ACA ad ATA are tested. This serological gap was the object of extensive research over the last decades. Anti-RNA polymerase III (RNAP3) autoantibodies were then associated to dSSc with rapidly progressive presentation, renal involvement, gastric antral venous ectasia (GAVE), and increased risk of cancer. Anti-RNAP3 autoantibodies were included in the 2013 ACR/EULAR classification criteria for SSc, along with ACAs and ATAs [3]. The relevance of autoantibodies in SSc is well established both for diagnosis and prognosis [4].

Patients negative for the conventional autoantibodies (ACA, ATA, and RNAP3) often present as an overlap syndrome or undifferentiated connective tissue disease (UCTD). The observation of nucleolar indirect immunofluorescence (IIF) ANA patterns on HEp-2 cells (ICAP AC-8, AC-9, and AC-10) lead to the description of anti-Pm/Scl, anti-fibrillarin, anti-Th/To, and anti-NOR-90 autoantibodies. A speckled ANA pattern (ICAP AC-4 and AC-5) is observed with anti-U1-RNP in mixed connective tissue disease (MCTD). Several other autoantibodies associated with MCTD-like or UCTD clinical features overlapping with SSc are associated with speckled patterns, such as anti-Ku, anti-U4/U6, and anti-U11/U12-RNP. A few nuclear dots pattern (AC-7) was recently described in association with anti-SMN autoantibodies in myositis/SSc overlap syndromes. Many other autoantibodies specific or associated to SSc have been described [5], such as anti-PDGFR [6], anti-BICD2 [7], anti-exosome [8], and anti-eIF2B autoantibodies [9]. The latter associate with a cytoplasmic fine-speckled HEp-2 IIF pattern [9]. Most of those autoantibodies are not routinely measured, and their use is limited to highly specialized laboratories or the research setting. This is due to the lack of standardization and studies validating their clinical role, which in turn leads to their unavailability in well-validated and commercially distributed assays.

Anti-RuvBL1 and anti-RuvBL2 are among the newest autoantibodies described in SSc patients. RuvBL1 (49 kD) and RuvBL2 (48 kD) are a nuclear complex involved in DNA repair and transcription [10]. They were originally identified by immunoprecipitation when two distinct bands of approximately 50 kDa were found [11]. Anti-RuvBL autoantibodies stained an ANA IIF in a fine-speckled pattern (AC-4) with increased fluorescence in the prophase and decreased in the metaphase [12]. Additionally, a fine-speckled pattern could be observed in the cytoplasm of HEp-2 cells in approximately 40% of positive sera [11]. Anti-RuvBL1/2 autoantibodies are highly specific for SSc, associated with PM/SSc overlaps with dSSc, and more frequently found in older patients of male sex [11,13,14] or, less frequently, in necrotizing polymyositis with morphea [13].

In this manuscript, we report a case of SSc with anti-RuvBL1 and anti-RuvBL2 autoantibodies identified using a novel particle multi analyte technology (PMAT) enabling the rapid detection of multiple SSc autoantibodies in a multi-analyte platform. We then reviewed the literature for papers describing patients with anti-RuvBL1/2 autoantibodies. 

## 2. Materials and Methods

Anti-RuvBL1/2 detection: ANAs were evaluated during routine clinical laboratory practice on a HEp2 cell substrate in IIF (Euroimmun, Lubeck, Germany), according to the manufacturer’s indications. The screening titer was 1:80, and end-titer dilutions were performed. 

The serum sample was tested with the CTD research panel based on PMAT (research use only, Inova Diagnostics), including antigens bound to paramagnetic particles (namely, dsDNA, RNP, Sm, Ro60, Ro52, SS-B, centromere, Scl-70, Jo-1, DFS70, Rib-P, Mi-2, TIF1-y, PL-12, SAE1, EJ, MDA5, HMGCR, PL7, SRP54, NXP2, Ku, RNA Pol III, Rpp38, Pm/Scl, BICD2, Rpp25, RUVBL1, RUVBL2, RNPC3, FHL1, Mup44, and PUF60). The PMAT technology (Werfen, San Diego, CA, USA, research use only) allows for the simultaneous detection of antibodies to various autoantigens. In brief, recombinant full-length proteins of RuvBL1 and RuvBL2 were coupled to paramagnetic particles that carry unique signatures and are incubated with diluted serum samples. After 9.5 min incubation at 37 °C, the particles were washed and incubated for 9.5 min at 37 °C with antihuman IgG conjugated to phycoerythrin (PE) to label the bound autoantibodies. After the final wash cycle, a median fluorescence intensity (MFI) on the particles was captured using a digital imager and analyzed using proprietary algorithms to derive meaningful information for each analyte. 

The cut-offs used to determine the prevalence were preliminary and based on screening disease controls (internal data). To verify the specificity of the RUVBL1 and RUVBL2 assays on PMAT, 253 disease control samples were tested, and a 99.2% specificity was obtained at the preliminary cut-off using a composition of controls. 

Review of the literature: A systematic search of Medline and Embase up to 30 November 2022 was performed using the medical subject heading terms “anti-RuvBL1/2”, “RuvBL1/2”, “systemic sclerosis”, “scleroderma”, “connective tissue disease”, “myositis”, and “idiopathic inflammatory myopathy” to identify publications. Manual searches of references cited in the retrieved articles were also performed. The eligibility criteria included (1) studies assessing anti-RuvBL1/2 antibodies in patients with systemic sclerosis; (2) studies assessing anti-RuvBL1/2 antibodies in patients with CTDs; (3) only peer-reviewed publications written in English and involving human subjects. Abstracts were excluded. No restriction on time was applied. Considering anti-RuvBL1/2-positive patients, the variables included prevalence among the recruited cohort, total number of cases, age at SSc onset, gender, organ involvement including pulmonary involvement (interstitial lung disease and/or arterial pulmonary hypertension), cutaneous involvement (Raynaud’s phenomenon, limited or diffuse involvement), gastrointestinal involvement (motility disorder), renal involvement, musculoskeletal involvement, cardiac involvement, neurologic involvement, ANA pattern on HEp-2 cells, other autoantibodies detected, and final diagnosis.

## 3. Results

### 3.1. Case Report

A 48 year old Caucasian male was referred to our outpatient clinic for a 3 year history of Raynaud’s phenomenon involving both hands and feet and recently worsened myalgias of the shoulder girdle and thighs with progressive muscle weakness.

His father died in his 70s from colorectal carcinoma, while his mother had hypertension and hyperthyroidism. He had a younger sister who was in good health. He had no children. He worked as a car mechanic and lived in a town in Tuscany, Italy.

At the age of 45, he had suffered from acute knee arthritis, and he had begun experiencing erectile dysfunction and Raynaud’s phenomenon involving both hands and feet. A year later, he was admitted to another hospital because of myopericarditis, and he was treated with NSAIDs, beta blocker, and ACE inhibitor. Because of elevated muscle enzymes and myalgias involving the shoulders and thighs, an electromyography was performed and showed inflammatory myopathy. The patient was treated with high-dose steroids and cyclophosphamide. Mycophenolate mofetil was added as maintenance treatment, with good clinical control.

When he was admitted to our clinic, he reported a 3 month history of shoulder and thigh pain, which was associated with muscle weakness and morning stiffness involving mainly the large joints. The Raynaud’s phenomenon was unchanged, and it appeared only during cold seasons. He reported no fever, cough, dyspnea, or chest pain. He denied dysphagia but sometimes he had to drink water to help solid food progression. No constipation or diarrhea were present. His medications included mycophenolate mofetil, methylprednisolone, bosentan, and sildenafil.

On physical examination, scleroderma facies with a paucity of wrinkles, thin lips, and sharp nose were observed (Figure 1A). The skin was mildly thickened (mRSS = 16), and there were telangiectasias on the malar area. He had swollen hands with Raynaud’s phenomenon occurring during the examination, but no digital ulcers or pitting were present (Figure 1B). Nailfold capillaroscopy revealed an early scleroderma pattern. Shoulder movements, especially adduction, were limited. Muscular strength was preserved. Cardiac, pulmonary, and abdominal examination were not notable.

Laboratory testing revealed mild microcytic anemia and lymphopenia, elevated C-reactive protein (CRP) (10 times the upper limit) and erythrocyte sedimentation rate (ESR) (3 times the upper limits). Muscle enzymes were mildly elevated. Renal and liver function were normal. An electromyography showed the myopathic involvement of both the upper and lower limbs. Signs of previous pericarditis, pulmonary hypertension (PAP: 45 mmHg), and mild atrial dilation with preserved ejection fraction (EF: 56%) were found upon echocardiography.

A high titer ANA (1:2560) with a fine-speckled pattern (ICAP AC-4) associated with a 1:320 cytoplasmic dense fine-speckled pattern (ICAP AC-19) was observed on HEp-2 cells (Euroimmun, Lubeck, Germany) (Figure 2A,B). An extended ENA CTD screening (FEIA, Phadia/Thermofisher, Uppsala, Sweden) and immunoblot (Euroimmun, Lubeck, Germany) testing for autoantibodies associated with SSc and myositis were negative.

The IIF pattern on HEp-2 cells was highly suggestive for anti-RuvBL1/2 autoantibodies. Therefore, the patient serum was analyzed using PMAT and tested positive for anti-RuvBL1 at 8292 MFI and for anti-RuvBL2 at 1119 MFI (mean fluorescence intensity, reference range < 100). 

A diagnosis of PM/Scl overlap was made, and the treatment was optimized by adding hydroxychloroquine to mycophenolate mofetil and methylprednisolone. Nevertheless, during the follow-up, inflammatory markers and muscle enzymes were steadily elevated, skin thickening worsened, and arthromyalgia was unchanged. Four months later, a pulmonary function test (PFT) revealed a moderate restrictive lung disease (FVC: 65%, FEV1: 68%, and TLC: 64% of the predicted value) and a severely reduced diffusing capacity for carbon monoxide (DLCO) (46% of the predicted value), even though the patient did not report cough or dyspnea. However, he did not live an active life because of muscular pain. A high-resolution CT scan of the chest revealed the initial signs of interstitial lung disease on both lung bases. A year after our first evaluation, the patient was started on rituximab 1000 mg in addition to the previous regimen of hydroxychloroquine and mycophenolate mofetil. After the induction (two doses two weeks apart), he reported a rapid improvement in muscular and articular pain, while muscle enzymes, ESR, and CRP gradually normalized. Electromyography normalized after 6 months from induction. Chest high-resolution computed tomography (HRCT) performed 12 months after the beginning of rituximab revealed no ILD progression. The forced vital capacity (FVC) and total lung capacity (TLC) were also significantly improved, but the DLCO was unchanged. No kidney involvement was detected upon urinalysis. Conversely, the patient developed mild dysphagia, and diffuse esophageal enlargement was revealed on esophagogram. As for skin involvement, it did not improve but slowly worsened. Rituximab therapy is currently ongoing at 1000 mg every 6 months. Autologous hematopoietic stem cell transplantation was already considered, especially if cutaneous involvement still severely progresses. Intravenous immunoglobulin (IVIG) therapy might be a further option to ameliorate the myopathy, skin thickening, and GI complications.

### 3.2. Review of the Literature

Fifty-one patients with anti-RuvBL1/2 antibodies have been reported in the English literature up to December 2022 [11,13,14,15,16,17] (Table 1). Twenty-eight cases were female and twenty-one male [11,13,14,15,16,17], with most patients between the fifth–sixth decade of life. Most of the reported cases (approximately 60%) were diagnosed with an overlap syndrome, mainly SSc/IIM (“scleromyositis”), SSc alone being the second most common diagnosis (approximately 35%). Of note, a case of SSc overlapping with Sjogren’s syndrome and one of SSc sine scleroderma overlapping with myositis have been described [14,17], as well as one uncommon morphea/myositis overlap [13]. Regarding organ involvement, myopathy aside, interstitial lung disease (ILD) and gastrointestinal conditions (e.g., gastroesophageal reflux disease and lower esophageal sphincter dysfunction) seem to be the most common manifestations, found in up to 88% and 94% of patients, respectively. Heart disease (including arrhythmias, bundle branch blocks, ischemic events, and heart failure) may involve up to 50% of subjects, while pulmonary arterial hypertension and severe renal impairment appear to be quite rare in this subset of SSc patients. In most reported cases (92%, 47/51), anti-RuvBL1/2 autoantibodies were isolated, while a few cases overlapping with other systemic sclerosis and/or myositis autoantibodies were found (RNAP3, Ku, and Th/T0). 

SSc and myopathies can be associated with cancer, especially with anti-RNAP3 and anti-TIF1-γ autoantibodies, respectively. Data from more patients and longer follow-up periods are needed to assess the correlation of anti-RuvBL1/2 autoantibodies with cancer. In the largest case series included in this review, only 1 patient out of 37 was diagnosed with cancer (ten months after anti-RuvBL1/2 detection) [11]. Our patient, at a three-year follow-up, showed no clinical or radiological evidence of neoplasms. 

Treatment response and follow-up data are scarce in the published literature. Therefore, the prognostic value of these autoantibodies is still unclear. Takahashi et al. achieved an improvement in the morphea and muscle weakness, along with the normalization of serum CK and aldolase levels, by treating their patient with oral prednisolone 50 mg/day and azathioprine 100 mg/day for two months (tapering not specified), with no subsequent relapse [13]. Nomura et al. reported treating one patient affected by scleromyositis with methylprednisolone pulse therapy, IVIG, oral prednisolone, and oral cyclosporine (dosages not available) with good response on the muscle symptoms, but with re-elevation of muscle enzymes during steroid tapering; the skin thickening was refractory to therapy [15].

## 4. Discussion

Our patient presented some of the typical features of SSc, i.e., RP, puffy fingers, severe cutaneous thickening, esophageal dysfunction, heart involvement, and ILD. Furthermore, myalgia was among the first symptoms reported, which led to laboratory and EMG findings demonstrating myositis. A diagnosis of SSc/IIM overlap was initially made. Based on the clinical features, the presence of a nuclear fine-speckled pattern associated with a cytoplasmic fine-speckled pattern on Hep-2 cells IIF prompted the consideration of anti-RuvBL1/2 antibodies, which were eventually detected. When using commercially available line immunoassays, a considerable proportion (almost 50%) of patients with scleromyositis have negative results [18,19]. This group of patients tends to develop SSc complications, malignancy, and has poor prognosis [19]. Novel SSc-related autoantibodies, such as anti-RuvBL1/2 or anti-SMN autoantibodies, not routinely tested could be implicated, as in the current case report. A careful investigation of the ANA pattern may provide a clue for the diagnosis.

Closing the serological gap in SSc is an ongoing challenge. Many autoantibodies end up in the “Death Valley”, but these “orphan” autoantibodies could indeed have a potential role in clinical diagnosis and patient stratification [20]. Identifying autoantibodies not included in the classification criteria has important implications. Firstly, they are useful for establishing a diagnosis, favoring an early diagnosis, when the clinical manifestations may still be undifferentiated, thus enabling a prompt treatment. Secondly, identifying clinical–serological syndromes helps to define the prognosis by predicting complications, such as cancer risk, and disease progression. These data have clinically actionable consequences, such as tailored screening or personalized treatments. Moreover, the use of autoantibodies as biomarkers to stratify disease cohorts will be pivotal for the design of future clinical trials. 

Currently, many patients with rare autoantibodies are either lumped together based on broad clinical diagnosis or lost during recruitment, because they fail to meet the inclusion criteria (e.g., non-anti-Jo-1 autoantibodies in myopathies). This leads to a vicious cycle, where these rare autoantibodies are not well known by clinicians, not tested by laboratories, and, therefore, not developed as commercial assays by the industry. 

To overcome this vicious cycle, the first step would be to have widely adoptable assays to test for those orphan autoantibodies. This will enable the identification and description of the clinical features, prognosis, and best therapies associated with orphan autoantibodies. 

The ever-increasing number of clinically relevant autoantibodies poses a practical and economical challenge for clinical diagnostic laboratories. The development of multi-analyte technologies is paving the way to the testing of large autoantibodies panels. The diffusion of commercial multiplex assays was at the root of the increase in interest and research in autoimmune myopathies. It is likely that several myositis-associated autoantibodies will be included in the next classification criteria that, so far, includes only anti-Jo-1 autoantibodies [21]. Similarly, we foresee that the next generation of multiplex assays with panels dedicated to SSc and UCTDs will result in the adoption of several orphan autoantibodies. In this report, we demonstrated the clinical utility of a novel PMAT assay for the detection of a broad myositis and SSc autoantibodies, enabling the detection of anti-RuvBL1 and anti-RuvBL2 autoantibodies.

## 5. Conclusions

Anti-RuvBL1/2 autoantibodies characterize a distinct SSc/myositis overlap phenotype with rapidly progressive cutaneous scleroderma, interstitial lung disease, and muscular involvement. This clinical–serological syndrome might be largely underestimated given the absence of these autoantibodies on current commercially available assays (as of December 2022). In a patient with SSc and/or myositis, the presence of a nuclear fine-speckled pattern (ICAP AC-4), particularly if associated with a cytoplasmic fine-speckled pattern (ICAP AC-19), on HEp-2 cells can hint to the presence of anti-RuvBL1/2 autoantibodies. Prospective studies are required to define the true prevalence, the complete phenotypical spectrum, and the outcomes of these patients.

## Figures and Tables

**Figure 1 antibodies-12-00013-f001:**
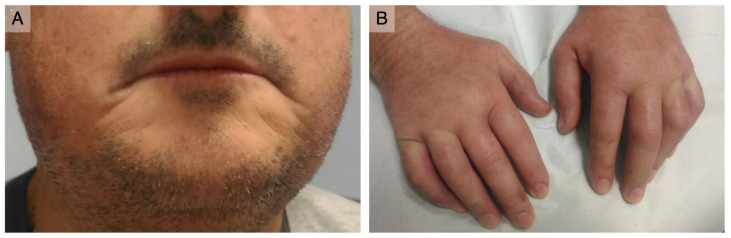
**Clinical presentation:** (**A**) thin and retracted lips; (**B**) puffy fingers with no digital ulcerations or scarring.

**Figure 2 antibodies-12-00013-f002:**
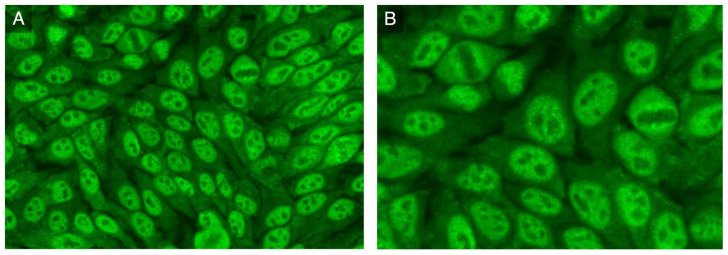
**HEp-2 indirect immunofluorescence:** (**A**) nuclear fine-speckled pattern and cytoplasmic dense fine-speckled pattern; (**B**) increased fluorescence in prophase and decreased in metaphase.

**Table 1 antibodies-12-00013-t001:** Main clinical and laboratory features of 52 anti-RuvBL1/2-positive SSc patients (up to December 2022).

Authors, Year [Ref.] (PMID)	No. of Patients	Age	Gender	Anti-RuvBL1/2 Autoabs Prevalence in the Cohort	Other SSc- or IIM-Related Autoabs	ANA Pattern on HEp-2 Cells IIF (Titer)	Organ Involvement	Final Diagnosis
Kaji et al. (2014) [11] (24023044)	Japan cohorts 10	58.1 ± 12.1	5 F 5 M	6/316 SSc pts. (1.9%) 4/272 SSc pts. (1.5%)	11% (4/37: 1 RNAP3/Ku, 1 RNAP3, 1 Ku, 1 Th/To)	Speckled (1:160–1:1280) + in 4/10 cytoplasmic granular	ILD 70%; GI 60%; Muscle 60%; Heart 50%; PAH 10%; Renal crisis 0%	dSSc/IIM 40%; dSSc 27%; lSSc/IIM 19%; lSSc 14%
Kaji et al. (2014) [11] (24023044)	Pittsburgh cohort 27	46.0 ± 15.1	17 F 10 M	27/485 SSc pts. (5.6%)		n/a	GI 94% *; Muscle 59%; ILD 50% *; PAH 13% *; Heart 22% *; Renal crisis 4%	
Takahashi et al. (2017) [13] (27786369)	1	42	1 F	-	SSc: Neg. IIM: AMA	Speckled (1:2560)	Neg	Morphea/IIM overlap
Pauling et al. (2018) [14] (29294089)	2	n/a	2 F	2/128 ANA neg. SSc pts. (1.6%)	SSc: Neg. IIM: n/a	ANA neg	PAH 50%; Muscle 50%; Renal crisis 0%; Others n/a	1 dSSc/IIM; 1 lSSc/SjS
Nomura et al. (2020) [15] (32031537)	1	21	1 F	-	Neg.	Speckled (1:1280)	Muscle; Lung; GI; Heart: no; Kidney: n/a	dSSc/IIM
Landon-Cardinal et al. (2020) [16] (32892170)	2	n/a	n/a	2/20 seronegative SSc pts. (10%)	Neg.	Fine speckled (1:640) 1/2; Large speckled (1:1280) 1/2	Muscle 100%; GI 100%; Renal crisis 50%; ILD 0%; Others n/a	dSSc/IIM
Vulsteke al. (2022) [17] (35027396)	8	52.6 ± 12	2 F 6 M	8/51 SSc pts. (16%)	Neg.	Fine or large speckled (≥1:80) + in 6/8 cytoplasmic speckled	ILD 88%; Muscle 62%; Heart 50%; GI 25%; PAH 0%; Renal crisis 0%	dSSc/IIM 25%; lSSc/IIM 25%; dSSc 25%; lSSc 12.5%; SSc *sine scleroderma*/IIM 12.5%
Di Pietro et al. (2022); Current case report	1	48	1 M	-	Neg.	Fine speckled + cytoplasmic speckled	Muscle, Heart, GI, ILD	SSc/IIM

Autoabs: autoantibodies; AMAs: antimitochondrial antibodies; dSSc: diffuse systemic sclerosis; F: female; GI: gastrointestinal tract; IIM: idiopathic inflammatory myopathy; ILD: interstitial lung disease; lSSc: limited cutaneous systemic sclerosis; M: male; n/a: not available; PAH: pulmonary arterial hypertension; Pts: patients; SjS: Sjögren’s syndrome. * Among patients for whom data are available.

## Data Availability

No new data were created or analyzed in this study. Data sharing is not applicable to this article.

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
