# Peer review of "Anti-RuvBL1/2 Autoantibodies Detection in a Patient with Overlap Systemic Sclerosis and Polymyositis"

_2073-4468, 2023, doi:10.3390/antib12010013_

Round 1

Reviewer 1 Report

This is an interesting paper reviewing anti-RuvBL1/2 autoantibodies with the presentation of a single case.

Although I agree that this appears to be a case of systemic sclerosis rather than dermatomyositis because of the oesophageal involvement and the absence of a rash, I still have some questions about the clinical aspects of the case.

Why was cyclophosphamide used for myositis? This would be very unusual. Why wasn’t IVIG used for the muscle disease. What is the data for IVIG for gut involvement?

Why was the patient on Bosentan and Sildenafil? Did he have pulmonary hypertension or was this simply for Raynaud’s?

What do the authors mean by an early scleroderma pattern on nailfolds?

What were the pulmonary pressures on ECHO?

The authors comment that the antibody findings altered the diagnosis and management. How was this the case in this patient who clearly had systemic sclerosis? What other diagnoses would the authors have entertained? I would argue that the finding of anti-RuvBL1/2 autoantibodies was of interest only in this case and did not alter treatment.

The authors argue that these antibodies are associated with an overlap disorder whereas the literature does not bear this out. It would appear that about half the patients with this antibody have limited disease and half have diffuse disease and that muscle involvement only occurs in about 60% of patients.

 Also, the authors make no mention of the prevalence of this antibody in patients without disease ie, rate of false positives.

As the authors are aware, RNAP3, anti-centromere and anti-topoisomerase are mutually exclusive. This does not appear to be the case with this antibody. A comment on this would improve the manuscript.

Reviewer 2 Report

Overall, it is a well written case report and brief review of the literature on the relatively recently described anti-RuvBL1/2 autoantibodies in SSc/PM overlap syndrome. 

My comments and questions are the following: 

1. Instead of the 'lung hypertension' term in the Introduction part the authors should use 'pulmonary hypertension' to be more appropriate.

2. There are some minor mispellings that should be checked, as 'diffused SSc' in the Introduction part. 

3. The authors should use consistently throughout the paper either 'dcSSc' or 'dSSc' to abbreviate the diffuse cutaneous SSc subset. 

4. The authors should also report whether they have discontinued or continued myophenolate mophetil, hydroxychloroquine and the glucocorticoid therapy after the introduction of rituximab. 

5. A word is missing from the sentence starting as 'Conversely, the patient developed mild dysphagia and marked (...), and diffuse esophageal..'. What was the marked symptom? 

6. Please comment in the 'Review of the literature' section on the suggested or used therapies found in the literature. 

7. Please comment on the prevalence of cancer in the presence of the anti-RuvBL1/2 Abs based on the available data. 

8. Please comment on the prognosis of disease  evolution in patients with these antibodies based on the available data. 
